# Low-Cost Water Quality Sensors for IoT: A Systematic Review

**DOI:** 10.3390/s23094424

**Published:** 2023-04-30

**Authors:** Edson Tavares de Camargo, Fabio Alexandre Spanhol, Juliano Scholz Slongo, Marcos Vinicius Rocha da Silva, Jaqueline Pazinato, Adriana Vechai de Lima Lobo, Fábio Rizental Coutinho, Felipe Walter Dafico Pfrimer, Cleber Antonio Lindino, Marcio Seiji Oyamada, Leila Droprinchinski Martins

**Affiliations:** 1Federal University of Technology—Parana (UTFPR), Toledo 85902-490, Brazil; faspanhol@utfpr.edu.br (F.A.S.); slongo@utfpr.edu.br (J.S.S.); msilva.2019@alunos.utfpr.edu.br (M.V.R.d.S.); jaquelinepazinato@gmail.com (J.P.); fabiocoutinho@utfpr.edu.br (F.R.C.); pfrimer@utfpr.edu.br (F.W.D.P.); 2Graduate Program in Computer Science, Western Paraná State University (UNIOESTE), Cascavel 85819-110, Brazil; marcio.oyamada@unioeste.br; 3Sanitation Company of Paraná (SANEPAR), Curitiba 80215-900, Brazil; adrianalobo@sanepar.com.br; 4Federal University of Parana (UFPR), Curitiba 80210-170, Brazil; 5Western Paraná State University (UNIOESTE), Toledo 85903-000, Brazil; cleber.lindino@unioeste.br; 6Federal University of Technology—Parana (UTFPR), Londrina 86036-370, Brazil; leilamartins@utfpr.edu.br

**Keywords:** low-cost sensor, water quality, Internet of Things, remote sensor, environmental monitoring, environmental measurements, remote water quality monitoring

## Abstract

In many countries, water quality monitoring is limited due to the high cost of logistics and professional equipment such as multiparametric probes. However, low-cost sensors integrated with the Internet of Things can enable real-time environmental monitoring networks, providing valuable water quality information to the public. To facilitate the widespread adoption of these sensors, it is crucial to identify which sensors can accurately measure key water quality parameters, their manufacturers, and their reliability in different environments. Although there is an increasing body of work utilizing low-cost water quality sensors, many questions remain unanswered. To address this issue, a systematic literature review was conducted to determine which low-cost sensors are being used for remote water quality monitoring. The results show that there are three primary vendors for the sensors used in the selected papers. Most sensors range in price from US$6.9 to US$169.00 but can cost up to US$500.00. While many papers suggest that low-cost sensors are suitable for water quality monitoring, few compare low-cost sensors to reference devices. Therefore, further research is necessary to determine the reliability and accuracy of low-cost sensors compared to professional devices.

## 1. Introduction

The concept of Internet of Things (IoT) is widely used in various sectors of society due to the proliferation and advancement of sensing and communication technologies [1,2]. Specialized electronic devices with minimal processing capabilities, also known as smart objects, are used in homes, industries, cities, large farms, and even small rural producers. These devices can measure and send monitored data to the Internet in real time, usually through a wireless communication network. Once the data is stored in the cloud, it opens up possibilities for data analysis, optimization, and real-time decision-making. The data collected is diverse and can include variables such as temperature, humidity, geographic location, heart rate, images of open or closed environments, and others. In the context of water quality monitoring, the data collected includes parameters such as potential of Hydrogen, Dissolved Oxygen (DO), turbidity, Oxidation-Reduction Potential (ORP), and others [3,4,5].

Continuous, remote, and reliable monitoring of water quality can improve the management and control of water quality. This, combined with the use of low-cost sensing devices, can increase public access to high-quality water. Worldwide, at least three billion people do not know the water quality they depend on because there is no monitoring [6]. Providing access to high-quality freshwater for the population and improving water quality in the surrounding area are ambitious goals set out in the United Nations 2030 Development Sustainable Agenda. Therefore, comprehensive and up-to-date water quality monitoring data is essential for decision-makers to ensure the availability and sustainable management of water resources for both human use and healthy aquatic ecosystems [7].

In Brazil, for instance, the National Water Agency (ANA) is the government agency responsible for managing and monitoring water resources. ANA operates a basic water quality network with 1340 monitoring sites throughout the country [8]. In this network, on-site analyses of four basic parameters, namely pH, dissolved oxygen, electrical conductivity, and temperature, are conducted using multiparametric probes during campaigns. However, monitoring is sporadic, and there are still large gaps in both spatial and temporal coverage of monitoring because Brazil is a continental country (0.16 points/1000 km${}^{2}$), and not all federal units can monitor the quality of their waters due to the high cost of multiparametric probes and the logistics involved.

While IoT applications for real-time water quality monitoring are expected to reduce costs associated with operations and logistics and increase the number of sites monitored, several challenges, such as availability, reliability, performance, scalability, interoperability, and security, need to be addressed. The perception or sensing layer is located at the bottom of the IoT model, and it contains the sensor responsible for monitoring water quality parameters in various water bodies such as rivers, lakes, reservoirs, and tanks. Ensuring the quality of the data, i.e., the reliability and accuracy of the measurements; is a main concern in this context [5,9,10]. Whereas reference instrument sensors are very expensive and based on proprietary technology, cost-effective sensors are also available in the market, known as low-cost sensors. For example, a single probe for the Hanna 9829 multiparameter probe costs around 1000 USD [11]. In contrast, low-cost sensors capable of measuring the same parameters are available for around 300 USD (see Table 4), which is less than a third of the value.

Low-cost sensors can be easily integrated into rapid prototyping boards such as Arduino and small development boards such as Raspberry PI. However, the lack of information about their characteristics, such as robustness, lifetime, and interference, among others, is a problem. Therefore, the following open questions arise: (i) To what extent are the measurements provided by low-cost sensors reliable to certify water quality? (ii) Which low-cost sensors are suitable for use in water quality monitoring systems for water resources?

The literature contains hundreds of papers that use sensors, electronic devices, and computer systems to monitor water quality in real-time for various purposes [12,13,14,15,16]. Despite this, there are currently no studies or databases documenting which low-cost sensors have been systematically used to monitor water quality, nor any evaluation of their performance compared to professional devices and their applicability in different environments on a continuous or sporadic basis. From a technological perspective, sensors are not standardized; each operates at different measurement frequencies, is made of different materials, and has different operating principles, power supplies, and data representations [5]. Furthermore, as far as we know, the existing literature reviews that address water quality and the Internet of Things also do not address low-cost sensors, including their manufacture, model, and cost.

In this sense, the objective of this paper is to conduct a systematic review of the literature to determine which low-cost sensors are being used for remote water quality monitoring and what performance has been achieved in making the measurements. We provide an overview of water quality low-cost sensors based on 10 guiding questions and point out the challenges and main recommendations to guide the practical application of this technology. We emphasize that while there are major challenges in operationalizing IoT applications, regardless of context, such as security fault tolerance [17,18], energy efficiency [19,20], and others. There are also many research opportunities, such as using machine learning to analyze water quality data [21,22]. However, this work focuses on the perception level of the IoT model, especially the low-cost sensor used for water quality monitoring.

The remainder of this paper is organized as follows. Section 2 presents the background on water quality parameters monitored in Brazil to determine the water quality index and briefly describes related review articles. Section 3 presents the water quality parameters monitored to determine the water quality index in Brazil and briefly describes related review articles. Section 4 presents the results obtained, and Section 5 concludes with the main challenges and recommendations.

## 2. Background

### 2.1. Water Quality Index

The water quality index is a tool that summarizes information about several physical, chemical, and biological parameters of water. It is a standardized and understandable way to inform the public and guide water quality planning and management. There is no consensus on the best index, which depends on the target (i.e., raw water for public supply, agriculture, aquaculture, water pollution, etc.) and the economic conditions of the country.

In Brazil, ANA is the entity legally responsible for implementing the National System for Water Resources Management System  [8] and uses the Water Quality Index (WQI) developed by the National Sanitation Foundation for the United States of America. Although the index has limitations, it is the primary one used in the country to infer water resource quality. The WQI-ANA consists of nine parameters, namely: DO, thermotolerant coliforms, the pH, Biological Oxygen Demand (BOD) at 20 ${}^{\circ }$C after 5 days, temperature, total nitrogen and phosphorus, turbidity and total solids [23]. These parameters are considered sufficient to determine water quality and classify water bodies. However, other parameters may help to conclude the characteristics of particular water resources, such as ORP and metals concentration.

The parameters of WQI-ANA are measured in situ with a multiparametric device and at the laboratory by taking water samples to analyze mainly BOD, coliforms, total solids, and total nitrogen and phosphorus. There is the development of multiparameter probes including BOD and coliform bacteria sensors [24,25,26], but there are yet challenges to validate them. The common practice is to measure these parameters in the water samples and, when feasible, also in other types of samples. Some most common basic parameters of water quality are briefly described in the following.

■DO: The main methods for measuring DO are the titrimetric, the electrochemical or the optical method [27]. The DO probes usually use optical and electrochemical (based on the oxidation-reduction reactions). There are two types of electrochemical probes, which can be polarographic or galvanic, while the optical probes are based on the extinction of luminescence in the presence of oxygen and, more recently, developed probes based on the fluorescence method. Wei et al. [27] mention in their review that polarography is currently the most widely used electrochemical method, with a simple structure, a wide range of applications, and mature technology. However, there are some problems, and they pointed out that oxygen sensors based on the fluorescence method could overcome these problems presented by electrochemical sensors. The manufacturer Atlas Scientific, for example, offers galvanic probes (consisting of a silicone membrane, an anode bathed in an electrolyte, and a cathode), while Vernier offers an optical option (using luminescence technology) and recommends its use for teaching.■ORP: This parameter is a measure of the electrical potential (electron exchange) in the semi-reactions of oxidation and reduction and indicates whether water (water with other substances) is either oxidized or reduced. It indicates the ability of water bodies to cleanse themselves, and, therefore, high values of ORP usually indicate high levels of oxygen in the water. The principle is electric, even considering the new alternative technologies, and due to the importance of the measurement that helps to understand the change of other parameters, ORP sensors are usually coupled in multiparameter probes [28].■pH: Parameters in commercially available multiparameter probes are measured using electrodes with different operating principles and probes structures. pH probes typically contain two electrodes (sensor and reference) that measure the potential of hydrogen or hydrogen activity in a solution. Common pH probes are glass electrodes—the membrane material that interacts with the sample that is pH sensitive—and use Ag/AgCl reference electrodes. The body of the probes can be made of many materials (glass, epoxy, polymer, etc.) with different resistance and durability. In addition, the junctions, which are porous connection points between the reference electrode and the sample, are another important sensitive feature of the pH probes, which gives them durability and application under different working conditions. Moreover, there are recent technologies (electrical and optical types) that use, for example, Ruthenium (Ru) and Titanium Dioxide (TiO${}_{2}$) in the sensors [28].■Temperature: Multiple processes occurring in the water are affected by temperature, affecting the concentration of parameters such as DO, pH, and conductivity, among others. Thus, the temperature sensor, like pH and conductivity, is a common sensor in multiparametric sensors/probes. According to Silva et al., the most common low-cost temperature measurement process is to use thermoelectric devices and/or resistive sensors [28].■Turbidity: Indicates the degree of attenuation experienced by a ray of light as it passes through water. This attenuation is due to the absorption and scattering of light by suspended matter (silt, sand, algae, soil residues, clay, etc.) and is, therefore, an optical principle. The measurement is an important parameter for determining water quality and in the operation of water treatment plants, as it affects the number of coagulants needed in the treatment process. Silva et al. [28] pointed out that recently a low-cost technology based on a nephelometric turbidity sensor has been developed to monitor water quality continuously and mentioned some papers presenting the development of turbidity sensors.

Finally, other important chemical parameters that are used to infer water quality are nitrogen (i.e., ammonia, nitrate), phosphorus, total solid in suspension, and biological parameters such as total coliforms, *Escherichia coli*, algae, and cyanobacteria. The multiparametric devices also provide other parameters (e.g., salinity, pressure, and conductivity) measured by sensors or calculated from a combination of measurements and displayed by the device.

### 2.2. Low-Cost Water Monitoring Sensors

The number of parameters measured, the robustness, accuracy, and sensitivity of the sensors, and the characteristics of the device (material, software, cable length, display, peripherals, etc.) are responsible for the cost. Low-cost sensors for water monitoring have emerged as a solution to improve and expand the monitoring system, especially in low and middle-income countries. There is no consensus or clear definition of low-cost water quality sensors in the literature. Usually, only the cost or price is considered, which ranges from 10 to 200 USD, according to Section 4.1. The cost and price depend on the development stage of the country and cannot be used exclusively as a reference. We believe that the definition of low-cost sensors for water quality monitoring should be based on the characteristics of the solution along with the cost. Open-source hardware and software, user-friendly interface, and easy integration with microcontrollers like Arduino or single-board computers, for instance, Raspberry Pi, are sure parameters that define a low-cost sensor.

In air quality monitoring, the criteria for defining low-cost sensors are more mature. The definition or understanding is that they are lower price, smaller size, and lower power sensors—a sensor class of non-regulatory technology that are generally easier to use and presented as turn-key, ready-to-deploy monitoring devices, making them attractive to researchers, educators, and the general public alike [29,30,31,32,33]. Although there are no performance standards for low-cost sensors set by the U.S. Environmental Protection Agency (EPA) or industry, the sensors are being widely deployed, including by the United Nations [34] in the Global Environment Monitoring System for Air, which is leading the deployment of affordable air quality monitoring networks to assess urban air pollution and is proactively exploring the feasibility of merging satellite and ground observations in developing countries where air quality data gaps have existed for decades.

### 2.3. Related Work

Most of the review articles published in recent years on low-cost water quality sensors and the Internet of Things (IoT) are related to IoT water monitoring systems, energy optimization, information, and communication technology systems for water resource monitoring, control, and management, implementation of smart and sustainable water resource management technologies, and finally, other more general articles mainly focused on water and air quality monitoring [19,20,22,35,36,37]. Moreover, not all of them have done a systematic bibliographic review, and most of them focus on IoT-based water quality monitoring systems without providing an overview of low-cost sensors for IoT water quality monitoring, their evaluation under laboratory and environmental conditions, and calibration protocols.

Akhter et al. [38] discusses the critical water parameters (temperature, pH, nitrate, phosphate, calcium, magnesium, and DO) for fisheries and reviews the sensors available to detect these parameters. The authors point out that most sensors are expensive, need improvements in sensitivity, power requirements, and IoT compatibility, and must be used continuously in the field. They propose a system called a low-cost system. They used commercial and home-built sensors, which they found to be low-cost and most effective for field applications in fisheries farming. They found that the performance of the sensors degraded with long-term use and proposed developing an algorithm for automatic calibration to solve this problem. However, they did not present results on accuracy (i.e., comparison to reference instruments) and did not address the other problems with the system when used in the field.

Petkovski et al. [39] conducts an SLR of IoT-based systems for aquaculture. The authors define five research questions to describe the types of sensors used, the types of single-board computers, the protocols for data transport, the cloud-based platforms used for aquaculture, and the benefits of IoT in aquaculture. They found seventeen types of sensors. Temperature, pH, and DO are the three most commonly used sensor types. Raspberry Pi, Arduino, and ESP are the most common single-board computers found. Despite describing the sensor types, the authors do not address their manufacture, sensor model, and sensor cost.

Ramírez-Moreno et al. [40] searched Scopus, Google Scholar, and IEEE Xplore databases for a review of sensors for sustainable smart cities. They selected 193 articles related to six categories related to smart cities (energy, health, mobility, security, water, and waste). Among smart city deployments in the world, only two are in South America and none in Africa, with most in Europe with recognized initiatives in Barcelona and Hong Kong. For water quality monitoring, only one initiative, according to results obtained by authors, should be increased in the last year. The paper did not provide an evaluation of commercially available water quality sensors, nor did it discuss the problems of low-cost IoT water quality monitoring in field applications. They recognized that future sensors need to be improved in terms of cost and energy consumption and also called for an increase in robustness.

Palermo et al. [41] presents an overview of smart technologies for water resource management, focusing on water management systems for conservation at the building level. They provide an overview of the key components used in smart water level monitoring systems, smart leak detection, and smart water consumption monitoring systems. At the same time, Manoj [42] presents a review of various Water Quality Monitoring Systems (WQMS) for fish ponds using the IoT solutions and implements a water quality management system using underwater sensors. The authors found fifteen papers (from the last decade) presenting WQMS, most of which use IoT and pH and temperature sensors. The cost of the sensors is not addressed, or they are considered low-cost and little information is given about the sensors used. Regarding sensors, it is interesting to highlight the use of nitrate (NO${}_{3}^{-}$ ) and ammonia (NH${}_{3}$, called AmmoLyt) sensors to measure concentrations in freshwater, which together account for the largest fraction of total nitrogen. Hangan et al. [35] published a summary of advanced techniques for monitoring and managing urban water infrastructures and pointed to the growing body of work related to water and information and communication technology systems.

Silva et al. [28] focus their review on advances in technological research for online and in situ water quality monitoring. They presented a summary of studies on new alternative technologies for monitoring color, temperature, DO, turbidity, chlorine, fluorine, metals, nitrogen, pH, phosphorus, ORP, algae and cyanobacteria, total coliforms and *Escherichia coli* parameters in water. They pointed out various works and technologies for measuring water quality parameters and that most are based on optical or electrochemical sensors. They recommended more robust analyses and assessments under real conditions due to the recent development of these technologies.

The increase in publications on water-related topics can be attributed to the significant rise in water demand, coupled with concerns about water quality, pollution, and population growth. To address these challenges, the utilization of sensors and monitoring systems has become essential for improving the monitoring of water resources in various domains. Table 1 provides an overview of the literature reviews referenced or described in this SLR. We describe 12 works, most of which were published in the last 2 years. Although we identified 24 articles related to this field in our SLR, none of them comprehensively discuss the utilization of low-cost sensors for water quality monitoring in the IoT context.

## 3. Methodology

A SLR is a study method that employs strategies to select, analyze, evaluate, and summarize papers from a database of documents on the topic to obtain a consistent investigation of the topic and even provide directions for future research. The research protocol selected for this study is based on the work of Barbara Kitchenham [43]. A summary of the protocol can be found in Figure 1. The SLR conducted in this work is based on the selection of texts that address the following main research question:■**RQ****: What low-cost sensors are being used for remote water quality monitoring?**

The survey included only texts published in English between the years 2019 to 2022 and available in the databases ACM Digital Library, IEEE Digital Library, MDPI, Science Direct, and Springer Link. Databases were selected based on the following criteria: (1) online search engine; (2) advanced search engine; (3) recognized basis. The keywords follow the search logic combination: “water quality” AND (“sensor” OR “sensors”) AND ((“low cost” OR “low-cost”) OR (“Internet of things” OR “IoT”)). During our study, we found research works that do not use the term “low cost” or “low-cost” but do use low-cost sensors to develop their applications according to the IoT concept. Therefore, “IoT” is used as a synonym for “low cost” in our work. Table 2 shows the search result in the selected databases. Initially, 3517 papers were found that contained the search string. Considering the immense number of works found in the Springer Link database and the impossibility of processing all of them manually, the selection process was automated. A script was developed in the Python language that selects only works whose titles and abstracts contain the previously defined string. After applying the script, a total of 127 works from the Springer Link database were selected for analysis. The online tool Parsifal [44] was used to select papers that met the research question of this systematic review. Parsifal is a software tool specifically developed to aid researchers in conducting systematic literature reviews. With its user-friendly interface, Parsifal offers many features and functions that streamline and automate the literature review process. It supports importing and exporting references from popular citation managers and fosters collaboration among multiple reviewers, making it a valuable tool for conducting rigorous and efficient literature reviews.

**Figure 1 sensors-23-04424-f001:**
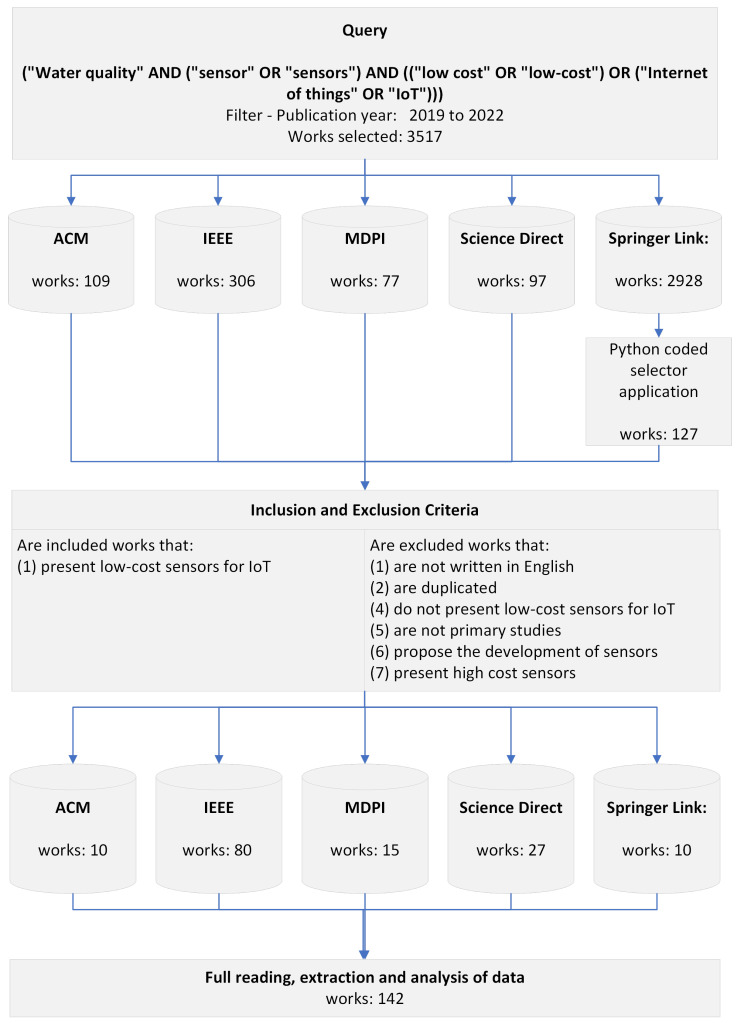
Systematic review protocol.

From this point on, the methodology followed three classification phases: (1) application of objective inclusion or exclusion criteria, (2) reading of titles and abstracts, and (3) full reading of the works. In the first phase, works before 2019, reviews (reviews or surveys), dissertations, theses, and speculative articles (towards new challenges, etc.), and those not written in English were excluded. In the second phase, articles whose titles and abstracts were related to the research question were selected. Finally, papers that used low-cost sensors for remote water quality monitoring were selected. In the third phase, the articles were read in full, and the 142 papers reported whether low-cost sensors were selected. Information on the total number of papers selected from each database of the repositories used as sources can be found in Table 2.

Based on the main research question, 10 additional research questions were elaborated to obtain detailed information about the study area. We would like to know not only which low-cost sensors were used to monitor water quality but also their cost, the environment in which the sensor was used, the result obtained and whether the results were compared with reference devices, the connection solution used, the country where the study was conducted, etc. The following 10 questions were defined to guide this SLR:■RQ1: What sensors were used, including their model, manufacturer brand, and cost?■RQ2: What water quality parameters were monitored?■RQ3: Did the sensors prove adequate considering their fields of application?■RQ4: Were the results obtained through low-cost sensors compared with the results of reference equipment (validation)?■RQ5: The sensors analyzed what environments (e.g., rivers, lakes, etc.)?■RQ6: Does the implemented solution have some connectivity to send data to the Internet in real time?■RQ7: In which country were the experiments realized?■RQ8: Has the number of citations increased in the years considered?■RQ9: What are the most cited studies?■RQ10: What are the limitations of the considered studies and the directions for future research?

## 4. Results

This section answers the research questions defined for this Systematic Literature Review (SLR). The terms *article*, *paper*, *study*, and *work* are used interchangeably and refer to the articles selected for analysis.

### 4.1. What Sensors Were Used?

Identifying low-cost sensors for water quality monitoring is one of the central issues of this SLR. Based on this, it is possible to create an initial database including the major sensor vendors, the technologies used to measure the physicochemical variables of interest, the cost/performance factor, etc. The identification of sensors will also allow the development of new works focused on analyzing the robustness, reliability, and durability of these devices when installed in different environments.

Table 3 lists the low-cost sensors used to monitor water quality, as identified in the selected articles. The majority of these sensors (46%) were manufactured by DFRobot, followed by Atlas Scientific (8%) and Vernier (2%). Some papers mention other manufacturers such as Thermo Fisher Scientific, Hach Company, Istek, Mettler Toledo, Asmik, Wilsen, Adafruit Industries, Daejin Instrument, Sensirion, HiLetGo, eKoPro, BHZY, Maxim Integrated, and Bosch Sensortec. However, it is worth noting that several reviewed papers do not mention the sensors used for their measurements; they only describe the measured parameters.

As reported by the manufacturer, Table 4 presents the cost, range, precision, and accuracy of sensors from the three most cited manufacturers (DFRobot, Atlas Scientific, and Vernier). The DS18B20 temperature sensor and the SEN0205 level sensor are the least expensive among the listed sensors, priced at US$6.90. Conversely, the ENV-50-DO sensor, which measures Dissolved Oxygen (DO), is the most expensive at US$ 353.99.

Among the cited manufacturers of the sensor, DFRobot stands out as the most frequently cited, as can be seen in Figure 2. The DFRobot sensors used in the reviewed articles include the following models: DO SEN0237, Electrical Conductivity (EC) DFR0300, Oxidation-Reduction Potential (ORP) SEN0165, potential of Hydrogen (pH) SEN0161 and SEN0169, Total Dissolved Solids (TDS) SEN0244, temperature sensor DS18B20, turbidity sensor SEN0189, and level/depth sensor SEN0205. These models are particularly noteworthy, given their frequency of use in the reviewed literature.

Although some works use only DFRobot sensors, as in the case of Concepcion et al. [45], Billah et al. [46], Ang et al. [47], others, such as Billah et al. [48], Abbasi et al. [49], opt for their association with sensors from other manufacturers. Following this strategy, Islam et al. [50] use DFRobot sensors to measure pH, TDS, and temperature, in conjunction with DO and BOD sensors from Hach Company and suspended solids sensors from Thermo Scientific. The association of DFRobot sensors with Atlas Scientific sensors is presented by Fonseca-Campos et al. [51]. Atlas Scientific manufactures environmental and electrochemical sensors used in environmental monitoring. The company offers a dedicated product line for IoT, which comprises three main models: env-20, env-40, and env-50. The env-20-DO is the smallest model. This sensor has a measurement range of 0–50 mg/L, a price of around US$135, and a life expectancy of 2.5 years. The env-40 model offers a higher measurement range of 0–100 mg/L for DO and has a life expectancy of approximately 4 years. It is priced at US$244 and is suitable for more demanding environmental monitoring applications. The env-50 is the industrial version of the DO sensor and is designed for heavy-duty industrial applications. It has a higher price point of around US$354, but it offers better performance and durability. To facilitate the integration of their sensors with microcontrollers and single-board computers, Atlas Scientific offers two boards, Gravity and EZO, which use analog, UART, and I2C communication protocols to interface with the sensors.

Madeo et al. [14] and Garuglieri et al. [52] use Vernier sensors and report accepted performance in measuring pH, ORP, DO, salinity, and flow in water quality monitoring in rivers, lakes, and coastal waters. The sensor models mentioned are PH-BTA, ORP-BTA, SALT-BTA, DO-BTA, FLO-BTA, and LJ-A. The cost of such sensors starts at US$127.00 for the pH sensor and increases to US$502.00 for the DO sensor. Arunplod [53], in turn, uses only Atlas Scientific sensors to measure temperature, pH, DO, and EC in rivers and lakes in the Philippines.

Furthermore, we highlight the use of some sensors whose technology is not owned by the previous manufacturers. This is the case with the DHT11, DHT12, DHT22, PT100, and LM35 temperature sensors that have been reported in works such as Abbasi et al. [49], Jayalakshmi and Hemalatha [54], Trevathan et al. [55]. Additionally, there are some sensors used in the literature whose manufacturers could not be identified, such as pH sensors E201-C, PH-4502C, DO D-6800, AR8210, and DOS-600, as well as turbidity sensors TSW-20M, BL5419, and ST100. These sensors were utilized in studies such as Simitha and Raj [56], Huan et al. [57], and AlMetwally et al. [58].

As for the processes required for reliable measurement results, all manufacturers establish specific calibration protocols for each sensor based on the technology used and the parameter measured. For instance, DFRobot offers calibration scripts based on the use of standard solutions and linearization processes, while Atlas Scientific offers scripts for calibrating its sensors using linearization processes with one to three points, depending on the parameter being measured.

The manufacturer Vernier also provides calibration protocols for its sensors, with each probe having its specific procedure. However, Vernier ties these protocols to the use of data acquisition equipment and proprietary applications provided by the manufacturer itself, such as LabQuest Mini, LabQuest 2 and 3, LoggerPro, and Graphical Analysis. This requirement can complicate the calibration process if such tools are not available.

In addition, the manufacturers recommend regular maintenance of the sensors depending on the characteristics of the water bodies under study. They point out that the sensors must be kept free of dirt deposits, as these can affect the functioning of the sensors and lead to measurement errors. It is also advisable to check the sealing systems to prevent water from entering the dry areas of the sensors, thus, ensuring their proper functioning. Section 4.3 will address whether the use of the low-cost sensors are considered adequate, from the point of view of the authors of each work, for their respective monitoring objectives. The various fields of application are comprehensively presented in Section 4.5, with a particular emphasis on monitoring water bodies, including rivers, lakes, seas, and springs, as well as aquaculture and fish farming applications.

### 4.2. What Are the Water Quality Parameters Monitored?

As described in Section 2, there are several parameters associated with water quality monitoring, the combination of which allows us to determine whether the water of a particular water body is suitable for certain critical uses, such as human consumption, animal consumption, agriculture, aquaculture, and so on.

Figure 3 shows the main parameters measured in the articles evaluated in this SLR. Several articles measured more than one parameter, and the figure summarizes the total number of articles by parameter. It can be observed that the parameter measured in a larger number of articles was pH (90%), followed by temperature (80%), turbidity (59%), DO (38%), and EC (36%). Of these parameters, only EC is considered a complementary parameter in the definition of water quality, as it is closely related to salinity and allows conclusions to be drawn about the content of dissolved salts in the water. Information on salinity and EC is crucial in classifying water as salty, brackish, or fresh, which helps define its use in regions with fresh and/or drinking water shortages.

Among the parameters shown in Figure 3, pH stands out clearly. This can be partly explained by the fact that several papers on aquaculture/hydroponics [48,49,57,59,60,61,62,63,64,65,66,67,68,69,70,71,72,73,74,75] and the pH is closely related to the metabolism of various aquatic species, so its monitoring and control is essential in this activity. Moreover, together with temperature, this is the sensor with more options of brands/models and technological maturity to do measurements in the aqueous medium. On the other hand, the DO content in water, a crucial parameter for maintaining aquatic life in a given water body, is less measured, which could be related to the application field (most on aquaculture/hydroponics) and the difficulty of performing reliable DO measurements in real conditions. Clean water generally has higher dissolved oxygen concentrations, about 5 mg/L, due to photosynthesis by algae and physical processes associated with water movement [8].

Finally, other parameters related to water quality were measured and mentioned less frequently in the articles reviewed: TDS, ORP, BOD, Fluorescent Dissolved Organic Matter (fDOM), suspended solids, total nitrogen, dissolved organic matter concentration, ammonia, nitrate, nitrite, iron, and magnesium, among others. In addition, about 13% of the studies mentioned parameters not directly related to water quality but to hydrological characteristics (water resource morphology). These include water level/depth, pressure, and flow.

### 4.3. Did the Sensors Prove Adequate Considering Their Fields of Application?

Most studies (111) considered that low-cost sensors were appropriate. Only 21 reported inadequate results, and 10 reported inconclusive results. However, most studies reported that they did not compare results with reference devices. See the next section for more details.

Among the reasons the authors gave for the unsatisfactory performance, those related to the need for maintenance and conservation stand out. In their work, Trevathan et al. [55] report damage to sensors due to water penetration into their dry parts. Wong et al. [76] report sensor malfunctions and measurement errors caused by the deposition of debris or biological encrustations on the sensors. Tholen et al. [77] reports on the degradation of electrical components by marine air. Technical problems are also mentioned, such as the lack of precision and accuracy of the sensors or their rapid degradation due to exposure to the environment in which they are installed [55,78,79,80].

In one recent paper, Xavier et al. [3] evaluated the DFRobot turbidity sensor SEN0189 compared to a reference device. The sensor withstood immersion and gave results close to those of the reference sensor during the 8-h test period. After this period, the difference between the response curves of the SEN0189 and the reference device increased continuously. At the time when the water change takes place (18 h), the error increases significantly. The sealing and the mechanical structure of the SEN0189 sensor are not prepared to be submerged for a long period of time.

### 4.4. Were the Results Obtained through Low-Cost Sensors Compared with the Results of Reference Equipment?

Although calibration with standard solutions is an important step to improve the reliability of low-cost sensors, it is not sufficient to guarantee their accuracy. It is essential to compare the results obtained from low-cost sensors with those of reference devices, such as multiparameter probes, to ensure their validity. This not only demonstrates the low-cost sensors’ proper operation but also helps to establish the accuracy and reliability of the obtained results. This is the focus of RQ4. In this regard, only 18 studies performed the expected comparison. On the other hand, 124 studies did not include reference instruments or did not discuss validation. In general, this lack of validation makes the results obtained with low-cost sensors less reliable.

Table 5 presents the works that bring a comparison of low-cost sensors with reference equipment. It is worth noting that the test periods in the reported studies were relatively short. Only one study [81] lasted for six months, while most studies were limited to a period of a few weeks or even a few hours. Among the papers comparing measurements of low-cost sensors with reference probes, the papers by Wu and Khan [82], Kinar and Brinkmann [83], Méndez-Barroso et al. [84], Bórquez López et al. [85] in which multiparametric probes from Xylem (YSI EXO, YSI 551, YSI-556) and Hanna Instruments (HI98128) are used as reference when testing pH, turbidity, TDS, DO, and temperature sensors. Probes from Eutech Instruments (CON 450) and Horiba are used as references by Weerasinghe et al. [86] and Demetillo et al. [15], respectively. In turn, other authors present the use of laboratory equipment or other sensors already installed in the test areas as reference sensors [51,57,65,87,88,89,90].

Wu and Khan [82] describe a novel seawater monitoring system that uses an unmanned floating surface vehicle in the form of a catamaran. The system is equipped with LoRaWAN transmitters, enabling remote data transmission and real-time monitoring of water quality parameters in seawater. The USV carries a suite of sensors from DFRobot to measure pH (SEN0169), turbidity (SEN0189), and temperature (DS18B20), as well as a DO sensor from Atlas Scientific, model not identified. These sensors were calibrated in the laboratory using the Labquest 2 instrument and Vernier sensors. In the field, the results were validated by comparison with measurements made with the multiparametric probe YSI EXO. Comparison data between the probe and the sensors are not described in the paper, nor is the evaluation time of the sensors.

To assess the water quality in coastal regions, estuaries, and tidal channels, Méndez-Barroso et al. [84] developed a monitoring system capable of measuring several parameters including water level, temperature, salinity, pH, EC, TDS, and DO in the water. The ENV40 sensors from Atlas Scientific were calibrated in the laboratory using standard solutions, and the results were compared with those obtained from the multiparameter Ysi Exo 3 probe. Field tests were conducted for 45 days. The results were validated with the Ysi Exo 3 probe and statistically evaluated using measures such as standard deviation, coefficient of determination ($R^{2}$), Root Mean Squared Error (RMSE), Pearson correlation coefficient, and statistical bias. From the analyses carried out, the authors report good performances of the low-cost sensors, emphasizing the accuracy and durability comparable to first-class equipment.

Moreover, for monitoring water quality in lakes, Huan et al. [57] propose an additional assessment of salinity by measuring the electrical conductivity of the water. In addition, the authors implement a correction of the pH, DO, and EC values measured with DFRobot sensors as a function of the temperature value [57]. Calibration laboratory procedures are performed before and after the field tests to evaluate the reliability of the results in terms of absolute error and mean square error over the 20 days of testing. The instruments used as a reference for calibration are an RTD sensor for temperature, and a FOPTOD ODO sensor for dissolved oxygen, while the pH and EC sensors were calibrated with standard solutions.

Standard pH and DO solutions were also used in the calibration procedures for Atlas Scientific sensors presented in Demethyllo et al. [15]. The results of the two-week tests, conducted in two streams, were validated by comparing the measured pH and DO values with those of a multiparametric probe from Horiba. The authors claim that the sensors perform well and evaluate the mean absolute error and coefficient of determination $R^{2}$ with values of 0.9792, 0.9731, and 0.9746 for DO, pH and temperature, respectively [15].

Bórquez López et al. [85] present a system for monitoring water quality in the context of precision aquaculture. In a laboratory setting, they implement and evaluate the performance of a low-cost open-source platform for measuring pH, DO, and temperature using Atlas Scientific sensors. Validation of the measured parameters was performed using Hanna HI98128 and ISY 155 multiparametric probes, obtaining coefficients of determination $R^{2}$ of 0.81, 0.72, and 0.97 for DO, pH and temperature, respectively [85]. The authors also evaluated the continuity of operation, reproducibility, and reliability. The continuity of the operation was evaluated by continuous monitoring of parameters for 30 days, and the results show proper system functioning. Reproducibility was confirmed by evaluating three similar systems and comparing the results obtained. Reliability, on the other hand, was confirmed by the survey or statistical analysis of the sensitivity, resolution, precision, and accuracy of the measurements.

In their study, Malissovas et al. [81] present a monitoring system for temperature, pH, and salinity (measured through EC) that is applied to rivers and water channels. The pH sensor’s raw data is corrected for instantaneous temperature, while the EC values are referenced to a temperature of 25 ${}^{\circ }$C using a linear compensation method. After temperature corrections and compensations, an algorithm based on phase angle measurement of impedance and voltage levels is used to analyze anomalous events, such as biofouling and possible sensor failures, to identify maintenance needs, repositioning, or replacement of the sensors. The authors evaluate the sensors’ performance for six months under adverse environmental conditions and without any maintenance interventions. Using unidentified reference sensors, they evaluate the low-cost sensors’ performance in terms of absolute and relative error, reporting correlations of 80% and 95% for pH and EC, respectively.

### 4.5. The Sensors Analyzed What Environments?

Sensors have been evaluated in a broad range of environments. In some cases, sensors are evaluated in strictly controlled laboratory settings (e.g., [51,95,96]), where environmental conditions are tightly regulated to provide consistent and reliable test conditions. In other papers, sensors are tested in more complex and dynamic environments, such as aquaculture tanks (e.g., [49,57,59,63,64,65,85,97,98,99]), rivers and lakes (e.g., [14,15,47,50,52,78,81,100,101,102,103,104,105]). These natural environments can present challenges that are not present in controlled laboratory environments, such as temperature variations, water currents, and the presence of contaminants. The intrinsic characteristics of the environments where sensors are evaluated can have a significant impact on their performance. For example, in natural aquatic environments, sensors may need to be designed to withstand biofouling, which is the accumulation of biological organisms on the sensor surface that can interfere with its functionality. Similarly, in industrial environments, sensors may need to be able to withstand exposure to harsh chemicals and extreme temperatures.

Figure 4 provides an overview of the number of papers that analyze sensor data in different environmental settings. The majority of the papers (56) focus on natural water bodies, with a specific emphasis on rivers (23), lakes and ponds (18), oceans, estuaries, and tidal channels (7), as well as aquifers and sources (8). Aquafarming applications (38) are the second most common area of focus. It should be noted that 16 articles did not provide information about the environmental context in which the sensors were tested.

### 4.6. Does the Implemented Solution Have Some Connectivity to Send Data to the Internet in Real Time?

Data collection is a crucial step in predicting future trends for all environmental indicators that define the conservation status of a given area. Therefore, connectivity is fundamental because it transmits the obtained measurements, whether in real time or not. However, having real-time information on water quality is essential for making decisions to protect public health, such as knowing the physicochemical properties of water. The connectivity solution to be chosen is directly related to the environment to be monitored. For example, long-range networks such as LoRa/LoRaWAN [106] are ideal for monitoring large geographic areas such as coastal regions, rivers, or lakes in both urban and rural environments. On the other hand, short-range solutions like Wi-Fi or Bluetooth are better suited for shorter distances within confined spaces.

Figure 5 summarizes the various connectivity solutions used in the studies. It is important to mention that some studies use multiple connectivity solutions. The most widely used connectivity solution is Wi-Fi/ IEEE 802.11, which accounts for almost 56% of the studies. The second most common category (17.6%) uses connectivity solutions based on mobile/cellular networks such as Global System for Mobile (GSM), 3G, 4G, General Packet Radio Service (GPRS), etc. A third significant proportion of studies (16.1%) uses Low-Power Wide-Area Network (LPWAN) connectivity, such as LoRa/LoRaWAN and Narrowband-IoT (NB-IoT). Several remaining connectivity options include local connections (7.7%), Bluetooth (6.3%), ZigBee (4.2%), and Radio Frequency (RF) (1%).

Figure 6 provides an overview of the platforms used in the studies. The most common platform is Arduino and variants such as Arduino + ESP32, Arduino + Raspberry Pi, and Arduino + ESP8266, which are used in 56% of the solutions. Arduino is used because it can be connected to different types of sensors, such as analog, standard digital interfaces (I2C, SPI), and custom digital interfaces. For Arduino solutions, connectivity is achieved using shields or communication boards for Wi-Fi, LoRaWan, Bluetooth, and Zigbee or by integrating the Arduino with the ESP8266 or ESP32. ESP32 alone is used in about 10% projects and has built-in Wi-Fi and Bluetooth connectivity. The Raspberry Pi (7%) is mainly used in solutions that require edge data processing. Platform solutions were not specified in about 10% of the selected studies.

### 4.7. In Which Country Were the Experiments Realized?

Table 6 summarizes the geographical distribution of the research, answering the RQ7. From this table, it can be observed that India, by far, corresponds to the highest number of studies (33). Bangladesh and Malaysia have the second position (11), followed by China and Indonesia (9) and Taiwan (7). Figure 7 displays the top-5 countries based on the number of studies.

The geographical distribution found, in part, reflects the higher demand for clean fresh water, the degree of water pollution bodies, and the income level and climate conditions of countries. For example, in 2017, 2 billion people worldwide did not have access to basic sanitation facilities such as toilets or latrines [107]. Another important aspect is that several articles analyzed fish farming and aquaculture environments, which are related to population food habits and food demand (China and India have the highest size population). Agriculture accounts for 70 percent of global water withdrawal according to the Food and Agriculture Organization of the United [108].

### 4.8. Has the Number of Papers Increased in the Period Considered?

The response to RQ8 is positive. During the time period considered in this SLR, it can be observed from Figure 8 that there was a consistent increase in the number of papers. Specifically, there were 19 articles in 2019, 25 in 2020, 34 in 2021, and a substantial rise to 64 in 2022. This suggests an interest of the research community in the use of low-cost sensors in water quality monitoring and control applications in different contexts. The low-cost sensors technology emerged as an alternative to standard sensors or devices, mainly because of cost and reduced size since there is an increased necessity to monitor continuously parameters of water quality that support the water uses and to quantify the impacts of human activities in the water bodies. As already mentioned, there are expressive inequalities in monitoring network systems among countries, and as pointed out by United Nations [109] we only manage what we measure.

### 4.9. What Are the Most Cited Studies?

Two papers stand out in the number of citations: Pasika and Gandla [110] (cited by 87) and Chowdury et al. [111] (cited by 83). However, most papers (112) have up to 5 citations, with an overall average of 5.9 citations.

Pasika and Gandla [110] propose a system for measuring pH, turbidity, and water level. In general, the contribution of the article is similar to the other reviewed works. It is noteworthy that the article describes in detail, in the form of diagrams, the interface algorithms between the sensors and the microcontroller devices for the acquisition and initial processing of the data. Moreover, the paper was published on an open-access platform, and the country in which it was developed was India, a country that accounts for about 23% of the publications reviewed in this paper (Table 6).

Published as open-access, the work by Chowdury et al. [111] presents a remote data collection platform for pH, temperature, turbidity, ORP, and electrical conductivity in rivers. A unique feature of this work is integrating the remote platform with a Big Data analysis system based on artificial neural network modeling. The measured parameters feed the neural network that produces an output to classify the water quality as either “good” or “bad”. Similarly to Pasika and Gandla’s work [110], Chowdury et al. [111] developed their work in Bangladesh, a region where the interest in developing devices for water quality analysis is quite significant.

In a paper published in 2020 in the journal Sensors (cited in 43 other works), Carminati et al. [112] proposed a novel approach for monitoring water quality parameters in the water distribution system of the city of Romagna, Italy. The authors combined off-the-shelf sensors with custom-built sensors to measure various parameters such as pH, temperature, flow rate, pressure, electrical conductivity, and thickness of the deposited biofilm on the inner walls of the pipes. The commercial sensors used in the study included the Sensorex S272CD for pH and temperature measurements, Digiten FL-808 for flow rate measurements, and Elco 3525VG1 for pressure measurements. The custom-built sensors were designed and implemented by the authors to measure the electrical conductivity and biofilm thickness deposited on the inner walls of the pipes. The performance of the slime film thickness measurement sensor, which is the authors’ main focus and a differential compared to other reviewed works, was tested in the laboratory, indicating a resolution of 10 μm in the measurement range of 10 to 300 μm. The performance of the measurement system was evaluated in terms of sensor resolution in continuous tests over two months. From the results, the authors report agreement with the state of the art and affirm that the apparatus exhibits satisfactory robustness.

### 4.10. What Are the Limitations of the Considered Studies and the Directions for Future Research?

Vernier, Atlas Scientific, and DFRobot are the main sensor suppliers for the validated studies. Vernier focuses on the education market, and some of their sensors have an integrated data transmission function. Atlas and DFRobot have focused their market on selling water-friendly sensors for the IoT concept. Atlas’ sensors have a modular design that allows for easy customization and integration with other devices, and their industrial version has similar features to DFRobot’s, highlighting its durability and longevity. DFRobot’s sensors have a compact size and low power consumption, making them ideal for remote monitoring applications. Both DFRobot and Atlas have versions for laboratory and industrial use, and their sensors have been used in a wide range of applications, such as water quality monitoring in aquaculture, hydroponics, and environmental monitoring.

In the evaluated studies, it is noticeable that sensors intended for laboratory use (e.g., Env-20 or SEN0161 for pH) are used when the sensor must be submerged for an extended period. Laboratory sensors are more fragile and not as well sealed as industrial sensors, which indicates that they are used in controlled environments and for sporadic readings. Industrial sensors are more expensive than laboratory sensors, although they still cost less than professional probes.

In any case, the evaluation made in most of the analyzed works is not about the lifetime of the sensor or the quality of the data obtained (e.g., accuracy, precision, sensitivity), whether in the laboratory or industry. SEN0169 is the most robust version of DFRobot for pH measurement, with SEN0169-v2 being its industrial version. The SEN0169 sensor is only used in three works, and none of them uses SEN0169-V2. The works that use SEN0169, like most other works, are about developing a system, an autonomous vehicle, evaluating the transmission medium, etc., but not about the sensor itself. In this sense, a significant limitation identified in this systematic review is the lack of evaluations of the sensor itself to confirm the quality assurance of the measurement results in a real environment. This becomes clear when considering that only 18 papers deal with evaluating their solution together with reference probes. There is also a gap in the procedures used and the statistical techniques applied.

We believe that progress in this area can only be achieved through studies that catalog low-cost sensors considering various characteristics such as measurement range, resolution, accuracy, precision, response time, reliability, lifetime, measurement method, and IoT solution. This should also take into account the potential sources of error or interference that could affect the performance of the sensor, such as cross-sensitivity to other analytes or environmental factors such as temperature, humidity, or pressure. Moreover, studies on a protocol for proper calibration, maintenance, and handling of low-cost sensors. Finally, studies that provide a robust comparison of low-cost sensors to certified sensors under various environmental conditions and perform quality assurance of low-cost sensor measurements. Although low-cost sensors could still be characterized for laboratory and industrial use, standardization of their properties concerning the reading environment could promote their widespread adoption.

To guarantee the optimal performance of a low-cost sensor, proper calibration, maintenance, and handling are also crucial. This may involve regular calibration against a reference instrument, careful storage and transportation, and periodic cleaning or replacement of components as needed. Users should also be aware of any software or firmware updates that may be necessary to ensure the sensor is functioning properly and providing accurate data. Overall, while low-cost sensors can be a valuable tool for many applications, users must take a thoughtful and thorough approach to sensor selection and maintenance to ensure the highest quality of data possible.

The turbidity sensor, which is present in 60% of the articles, can only be found in the laboratory version of DFRobot. This means that without modifications to the sensor to make it more robust and resistant to water penetration into the electronic parts, it would not be possible to obtain long-term real-time measurements for this parameter. This highlights the need for further research to develop reliable low-cost sensors. There is certainly much work to be done both in the search for materials and in the search for principles that will allow sensors to be improved and their cost reduced without sacrificing reliability. From a computational point of view, it is also possible to invest in computational algorithms and techniques for failure detection and error correction, as in the work of [81], and calibration algorithms such as those based on machine learning [113]. There are also water quality parameters for which there are no low-cost commercially available sensors on the market, as in the case of organic pollutants.

Another problem is the methodology used in the reviewed studies: few of them describe the sensor used, the software used to read it, and the calibration performed. Additionally, the test duration is relatively short. As mentioned earlier, this is because most of the studies do not focus on sensor evaluation. With this in mind, although the IoT concept makes it possible to measure water quality in previously unimaginable locations, efforts still need to be made to increase the robustness of low-cost sensors.

## 5. Conclusions

This SLR provides an overview of scientific papers dealing with low-cost sensors for water quality monitoring. Information is lacking on the accuracy, precision, durability, calibration procedure, and reliability of low-cost sensors needed for specific applications. Most of the work uses commercial sensors such as those from Atlas Scientific, Vernier, and DfRobot, which manufacturers claim are suitable for laboratory, educational, industrial, and residential purposes. However, only some works have made comparisons with reference devices. Therefore, it is important for users to thoroughly evaluate the capabilities and limitations of the sensor to ensure that it meets the data quality requirements for the intended application. This is because although low-cost sensors can be an attractive option due to their low price, they do not always provide the accuracy, precision, and reliability required for specific applications.

In this sense, our recommendations for future work are (i) long-term studies for multiple comparisons of low-cost sensors; (ii) experimental studies for field use with low-cost sensors and reference instruments; (iii) studies that support the development of a protocol for calibration of low-cost water sensors that archives requirements for environmental applications; (iv) new development of low-cost sensors, especially for turbidity due to the limited number of branches found in this SLR, and also for other important water quality parameters such as BOD, metals, and organic chloride; (v) Further studies under real conditions with remote and continuous water quality monitoring.

Finally, although a systematic review of the leading research databases was undertaken in this paper, the likelihood that some articles may not have been recorded by the authors during the selection phase cannot be ruled out. Another limitation is that this SLR doesn’t include articles in languages other than English.

## Figures and Tables

**Figure 2 sensors-23-04424-f002:**
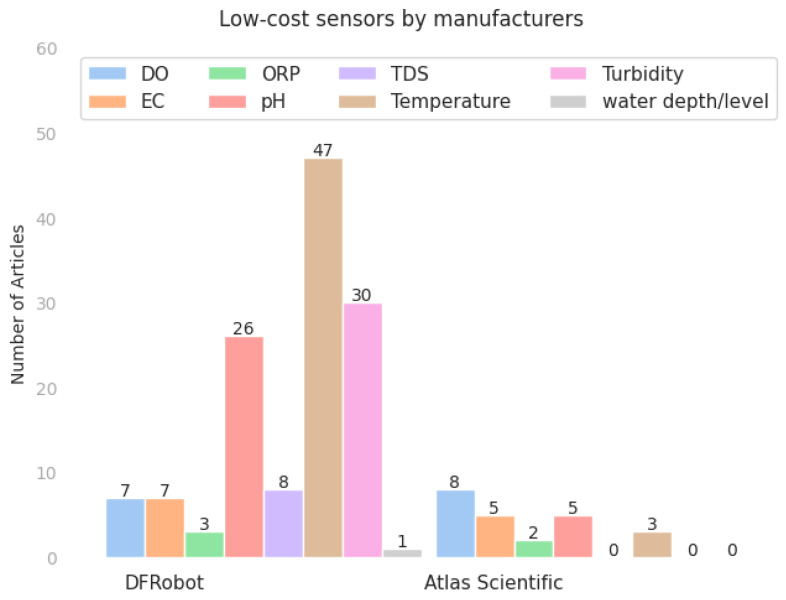
Low-cost sensors from the two most frequently cited manufacturers.

**Figure 3 sensors-23-04424-f003:**
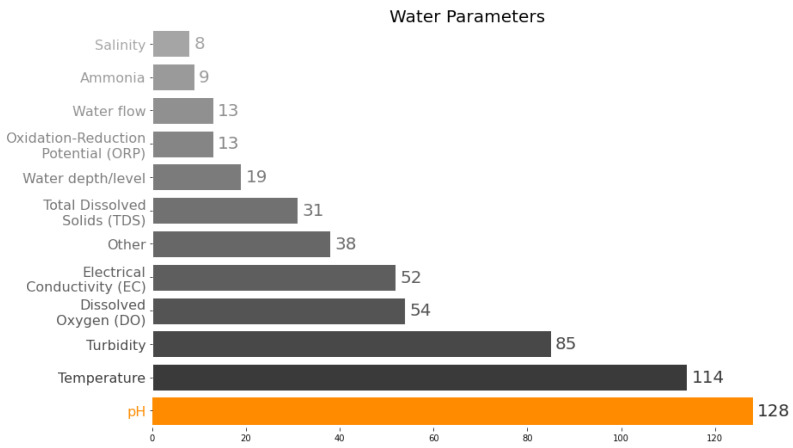
Water quality parameters measured.

**Figure 4 sensors-23-04424-f004:**
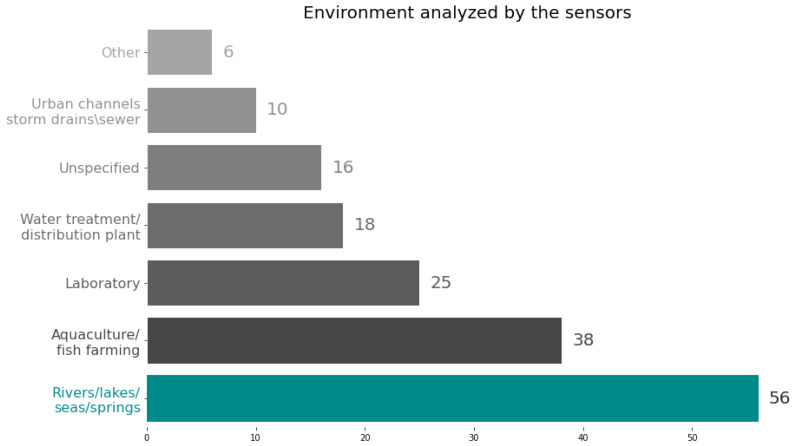
Number of papers reviewed considering the environment analyzed by sensors.

**Figure 5 sensors-23-04424-f005:**
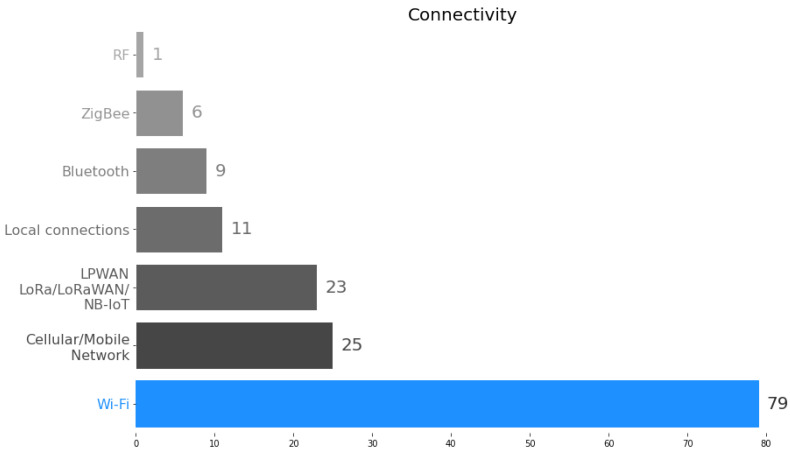
Connectivity categories.

**Figure 6 sensors-23-04424-f006:**
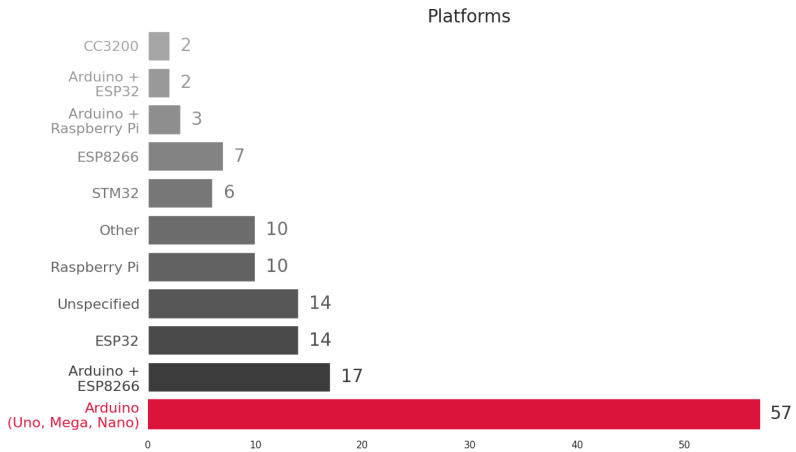
Most frequently cited platforms.

**Figure 7 sensors-23-04424-f007:**
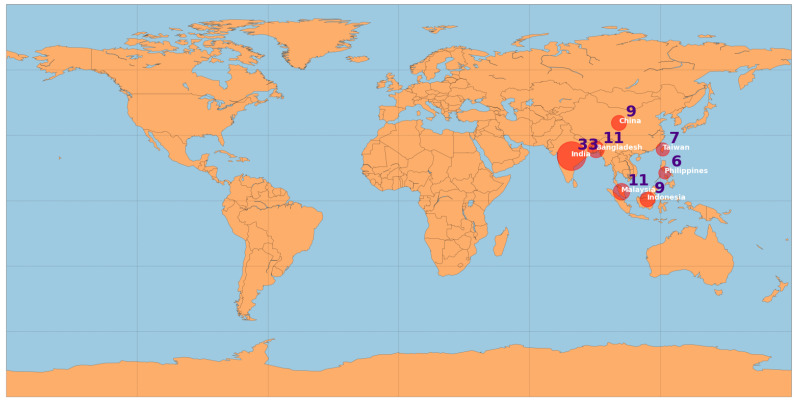
Geographic distribution of the studies: Top-5 countries.

**Figure 8 sensors-23-04424-f008:**
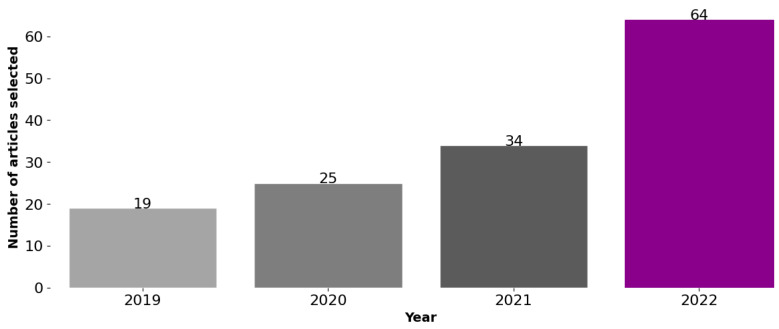
Number of articles published since 2019 selected to this SLR.

**Table 1 sensors-23-04424-t001:** Summary of literature reviews related with water quality and IoT.

Work	Title	Summary
Akhter et al. [38]	Recent Advancement of the Sensors for Monitoring the Water Quality Parameters in Smart Fisheries Farming	The research discusses the critical water parameters for fisheries and reviews the existing sensors to detect those parameters.
Hangan et al. [35]	Advanced Techniques for Monitoring and Management of Urban Water Infrastructures—An Overview	Conduct a review to show how emerging technologies offer support for smart administration of water infrastructures
Kesari Mary et al. [19]	Energy Optimization Techniques in Underwater Internet of Things (UIoT): Issues, State-of-the-Art, and Future Directions	Provides a survey on battery optimization issues in UIoT.
Manoj et al. [42]	State of the Art Techniques for Water Quality Monitoring Systems for Fish Ponds Using IoT and Underwater Sensors: A Review	Provides a summary of existing systems including technology, board and monitored water parameters.
Olatinwo and Joubert [20]	Energy Efficient Solutions in Wireless Sensor Systems for Water Quality Monitoring: A Review	Presents energy-efficient solutions for wireless sensor systems intended for the monitoring of water quality at water stations.
Palermo et al. [41]	Smart Technologies for Water Resource Management: An Overview	This work reviews smart and sustainable technologies for water resource management, primarily for building-scale uses.
Petkovski et al. [39]	IoT-based Solutions in Aquaculture: A Systematic Literature Review	This paper is a systematic literature review about the IoT-based applications in aquaculture.
Ramírez-Moreno et al. [40]	Sensors for Sustainable Smart Cities: A Review	This review presents an analysis of different sensors that are typically used in efforts toward creating smart cities in the field of energy, health, mobility, security, water, and waste management.
Silva et al. [28]	Advances in Technological Research for Online and In Situ Water Quality Monitoring—A Review	The paper review the development of modern technologies aimed at monitoring water quality, with the ability to reduce the costs of analysis and accelerate the achievement of results for management and decision-making.
Ubina and Cheng [36]	A Review of Unmanned System Technologies with Its Application to Aquaculture Farm Monitoring and Management	Conduct a review to provide an overview of the capabilities of unmanned systems to monitor and manage aquaculture farms that support precision aquaculture using the IoT.
Ullo and Sinha [22]	Advances in Smart Environment Monitoring Systems Using IoT and Sensors	Review on Smart Environment Monitoring (SEM) systems that involve monitoring of air quality, water quality, radiation pollution, and agriculture systems. The authors also describe the sensors used, the machine-learning techniques involved, and the classification methods found in each.
Zulkifli et al. [37]	IoT-Based Water Monitoring Systems: A Systematic Review	Search what kinds of data acquisition system (DAS) are now employed to gather water samples for testing and monitoring.

**Table 2 sensors-23-04424-t002:** Overview of studies number considered in this SLR.

Source	Initial Quantity	Selected	Accepted
ACM Digital Library	109	109	10
IEEE Digital Library	306	306	80
MDPI	77	77	15
Science Direct	97	97	27
Springer Link	2928	127	10

**Table 3 sensors-23-04424-t003:** Low-cost sensors used to monitor water quality parameters.

Supplier	Sensor	Articles
DFRobot	DO	7
	electrical conductivity	7
	ORP	3
	pH	26
	TDS	8
	temperature	47
	turbidity	30
	water depth/level	1
Atlas Scientific	DO	8
	electrical conductivity	5
	ORP	2
	pH	5
	temperature	3
Vernier	DO	1
	ORP	1
	pH	1
	electrical conductivity	1
	salinity	1
	water flow	1
Hach Company	DO	1
	BOD	1
Thermo Scientific	suspended solids	1
Other manufacturers	–	44
Not identified	–	77

**Table 4 sensors-23-04424-t004:** Low-cost sensors from three main manufacturers. FS stands for *Full-scale reading*.

Manufacturer	Parameter	Sensor Model	Cost (US$)	Range	Precision/Accuracy
DFRobot	DO	SEN0237	169.00	0∼20 mg/L	–
	EC	DFR0300	69.90	0∼20 ms/cm	±5% FS
	ORP	SEN0165	89.00	−2 V∼2 V	±10 mv (25 ${}^{\circ }$C)
	pH	SEN0161	29.50	0∼14 pH	±0.1 pH (25 ${}^{\circ }$C)
	pH	SEN0169	56.90	0∼14 pH	±0.1 pH (25 ${}^{\circ }$C)
	TDS	SEN0244	11.80	0∼1000 ppm	±10 % FS (25 ${}^{\circ }$C)
	temperature	DS18B20	6.90	−55∼125 ${}^{\circ }$C	±0.5 ${}^{\circ }$C (−10∼85 ${}^{\circ }$C)
	turbidity	SEN0189	9.90	–	–
	water depth/level	SEN0205	6.90	–	±0.5 mm
Atlas Scientific	DO	ENV-20-DOX	134.99	0∼50 mg/L	±0.2 mg/L
	DO	ENV-40-DOX	214.99	0∼100 mg/L	±0.05 mg/L
	DO	ENV-50-DO	353.99	0∼100 mg/L	±0.05 mg/L
	EC	ENV-20-EC-K1.0	123.99	5∼200,000 μS/cm	±2 %
	EC	ENV-40-EC-K1.0	157.99	5∼200,000 μS/cm	±2 %
	EC	ENV-50-EC-K1.0	274.99	5∼200,000 μS/cm	±2 %
	ORP	ENV-10-ORP	243.99	−2∼2 V	±1 mV
	ORP	ENV-20-ORP	83.99	−2∼2 V	±1 mV
	ORP	ENV-30-ORP	58.99	−1.1∼1.1 V	±1.1 mV
	ORP	ENV-40-ORP	128.99	−2∼2 V	±1 mV
	ORP	ENV-50-ORP	237.99	−2∼2 V	±1 mV
	pH	ENV-10-pH	237.99	0∼14 pH	±0.002 pH
	pH	ENV-20-pH	60.99	0∼14 pH	±0.002 pH
	pH	ENV-30-pH	48.99	2∼13 pH	±0.1 pH
	pH	ENV-40-pH	85.99	0∼14 pH	±0.002 pH
	pH	ENV-45-pH	139.99	0∼14 pH	±0.002 pH
	pH	ENV-50-pH	234.99	0∼14 pH	±0.002 pH
	temperature	ENV-10-TMP	64.99	−200∼200 ${}^{\circ }$C	±(0.15 + 0.002 ×T)
	temperature	ENV-50-TMP	70.99	−55∼220 ${}^{\circ }$C	±(0.15 + 0.002 ×T)
Vernier	DO	DO-BTA	–	0∼15 mg/L	±0.2 mg/L
	DO	ODO-BTA	–	0∼20 mg/L	±0.2 mg/L
	DO	GDX-ODO	–	0∼20 mg/L	±0.2 mg/L
	EC	GDX-CONPT	–	0∼20,000 μS/cm	±10 μS/cm
	EC	GDX-CON	–	0∼20,000 μS/cm	±1 % FS
	ORP	GDX-ORP	–	−1∼1 V	±20 mV
	ORP	ORP-BTA	–	−450∼1100 mV	–
	pH	GDX-PH	–	0∼14 pH	±0.2 pH
	pH	PH-BTA	–	0∼14 pH	±0.2 pH
	salinity	SAL-BTA	–	0∼50,000 ppm	±1 % FS
	flow velocity	FLO-BTA	–	0∼4.0 m/s	±1 % FS

**Table 5 sensors-23-04424-t005:** Comparison of low-cost sensors with reference equipment. NSS stands for *Non-identified Standard Sensors*.

Work	Standard Equipment	Testing Period	Statistical Analysis of Data
Adriman et al. [61]	Refractometer Atago and PCS Tester 35	Not informed	Relative error
Bórquez López et al. [85]	HANNA HI98128 and YSI 551 multiparameter probes	5 days	$R^{2}$, variance, mean, standard deviation, etc.
Demetillo et al. [15]	Horiba Water Checker	Not informed	$R^{2}$ and absolute error
Goparaju et al. [91]	HANNA HI 99300 multiparameter probe	Not informed	$R^{2}$ and root mean squared error (RMSE)
Hawari and Hazwan [65]	NSS	13 h	Standard deviation and absolute error
Huan et al. [57]	HASH COMPANY MS5—Hydrolab	Not informed	relative error
Kinar and Brinkmann [83]	YSI EXO2 multiparameter probe	Not informed	None
Malissovas et al. [81]	NSS	6 months	Relative error and absolute error
Martínez et al. [92]	NSS	1 month	$R^{2}$, Relative error and standard deviation
Méndez-Barroso et al. [84]	YSI EXO3 multiparameter probe	3 months	$R^{2}$, RMSE, standard deviation, Pearson correlation coefficient and bias
Nandakumar et al. [88]	NSS	Not informed	Relative error
Rezwan et al. [87]	NSS	1 day	None
Singh et al. [90]	Systronics 802 pH meter	Not informed	None
Tsai et al. [93]	NSS	20 days	None
Wannee and Samanchuen [89]	NSS	30 min	Relative error
Weerasinghe et al. [86]	Thermo Scientific Eutech CON 450 and other NSS	Not informed	RMSE
Wu and Khan [82]	YSI EXO multiparameter probe and Vernier sensors	Not informed	None
Xu et al. [94]	CHI660E electrochemical workstation	Not informed	$R^{2}$ and relative error

**Table 6 sensors-23-04424-t006:** Geographic distribution of the studies.

Country	Articles	Country	Articles	Country	Articles
India	33	Brunei	2	Morocco	1
Bangladesh	11	Canada	2	Netherlands	1
Malaysia	11	Cyprus	2	New Zealand	1
China	9	Egypt	2	Nigeria	1
Indonesia	9	Japan	2	Pakistan	1
Taiwan	7	Peru	2	Portugal	1
Philippines	6	Saudi Arabia	2	Russia	1
USA	5	South Africa	2	Senegal	1
Australia	4	Brazil	1	Sudan	1
Italy	4	Ecuador	1	United Kingdom	1
Mexico	3	Fiji	1		
South Korea	3	Iraq	1		
Spain	3	Jordan	1		

## Data Availability

Not applicable.

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
