# Peer review of "Low-Cost Water Quality Sensors for IoT: A Systematic Review"

_sensors, 2023, doi:10.3390/s23094424_

Round 1
Reviewer 1 Report
The paper conducts a systematic literature review to determine which low-cost sensors are being used for remote water quality monitoring and what performance has been achieved in making the measurements. In general, the paper is well-written, and the systematic literature review protocol seems consistent.
- In the question "RQ1: What sensors were used, including their model, manufacturer brand, and cost?" (or maybe RQ2) a valuable discussion would be which applications used the sensors (irrigation, urban monitoring, among others)
- Regarding RQ4: the sensor precision is a key parameter that directly impacts the sensor cost. In general, commercial sensors have this information in their documentation (a % of precision error), or the results of the tests in the papers can be discussed. This discussion and information about sensors presented should appear in this survey.
- Regarding RQ6: the connectivity is a parameter of the solution (microcontroller and communication standard) used. Discussing also the microcontroller (or protoboard) used in the analyzed works (and maybe compatible with the specific sensor analyzed) is an interesting contribution.
- RQ7, RQ8 and RQ9 do seem not to be research questions but analyses of their results (initial bibliometrics). Regarding RQ9, the impact of a paper can not be evaluated just for how many times it was cited. The authors should change this term in that discussion.
- Tables must be included in some parts of this paper. It can help to visualize some results of RQs more easily.
Reviewer 2 Report
Summary/Contributions: A systematic literature review was undertaken in this study to establish which low-cost sensors are employed for remote water quality monitoring. According to the findings, there are three key vendors for the sensors utilized in the selected publications.
Comments/Suggestions: 1. This survey is well-written and covers an interesting topic. 2. The authors are invited to add a short paragraph about related surveys dealing with the same topic in order to emphasize the originality of their approach. 3. Since this paper is a systematic literature review, it is necessary to provide more details about the adopted search methodology (like inclusion, exclusion criteria, papers collected by the snowball effect, etc.) 4. The authors are invited to enrich the paper with more tables and figures that summarize the main findings of the different sections. This will make the paper easier to read and the reported results easier for readers to memorize. 5. The authors may add a paragraph that deals with the use of well-known formal methods for the protection of IoT systems from security breaches 6. For this purpose, they may include the following interesting papers (and others):a. https://link.springer.com/chapter/10.1007/978-3-030-13705-2_26 b. https://link.springer.com/chapter/10.1007/978-3-030-02807-7_9 7. Line 494: "Bórquez López et al. present a system for monitoring" ===> please insert the corresponding reference. 8. Please avoid the use of very short paragraphs everywhere in the paper. 9. Is it possible to use built-in smartphone sensors for the same purposes considered in this work. 10. The authors are invited to identify the limitations of their study and propose more future work directions.
Reviewer 3 Report
First of all, I want to congratulate the authors for their efforts in the manuscript. They had done a hard job searching and analyzing the different papers about water quality. The topics are completely aligned with the journal's scope and are interesting for the readers. There are some issues to be addressed before accepting the paper. Following, I include a series of comments aimed at enhancing the quality of their survey:
It is not clear if Table I corresponds to only the first question of the summary of all questions. If it corresponds to the first question, the authors must provide the tables for the rest of the questions.
In Section 3, please clarify the search tool used.
The following papers should be included in the survey since it proposes the use of low-cost sensors for water quality monitoring in a LoRa network (https://doi.org/10.1007/s11036-022-01994-8).
Consider using a graphic representing the number of papers per country in the globe as an alternative for Table 5.
At the end of the results, it is recommended to add a new section in which the current challenges and future trends are identified by the authors.
Conclusions should be reduced if possible.
Minor issues:
In the keywords, avoid using the terms already included in the title
Related works should be related work
Round 2
Reviewer 2 Report
The authors considered all my comments and suggestions. Good luck.
Reviewer 3 Report
The authors have improved the paper and now it is ready to be accepted.